# Association between Fruit Consumption and Lipid Profile among Children and Adolescents: A National Cross-Sectional Study in China

**DOI:** 10.3390/nu14010063

**Published:** 2021-12-24

**Authors:** Jieyu Liu, Yanhui Li, Xinxin Wang, Di Gao, Li Chen, Manman Chen, Tao Ma, Qi Ma, Ying Ma, Yi Zhang, Jun Jiang, Zhiyong Zou, Xijie Wang, Yanhui Dong, Jun Ma

**Affiliations:** 1Institute of Child and Adolescent Health, School of Public Health, Peking University, Beijing 100191, China; jieyulynne@163.com (J.L.); yanhui_lyh@163.com (Y.L.); gaodi1993@163.com (D.G.); clcl@bjmu.edu.cn (L.C.); 1911210173@pku.edu.cn (M.C.); 1610306216@pku.edu.cn (T.M.); 18702110295@163.com (Q.M.); mypku232@163.com (Y.M.); 1710306140@pku.edu.cn (Y.Z.); harveyzou2002@bjmu.edu.cn (Z.Z.); majunt@bjmu.edu.cn (J.M.); 2National Health Commission Key Laboratory of Reproductive Health, Peking University, Beijing 100191, China; 3School of Public Health and Management, Ningxia Medical University, Yinchuan 750004, China; wangxinxin291314@163.com; 4Department of Plant Science and Landscape Architecture, University of Maryland, College Park, MD 20742, USA; sherryj@umd.edu; 5Vanke School of Public Health, Tsinghua University, Beijing 100084, China; 6Institute for Healthy China, Tsinghua University, Beijing 100084, China

**Keywords:** fruit consumption, lipid, dyslipidemia, children and adolescents, China

## Abstract

To investigate associations between fruit consumption and lipid profiles, and to further explore a satisfactory level of frequency and daily fruit intake for children and adolescents. A national sample of 14,755 children and adolescents aged 5–19 years from seven provinces in China were recruited. Fasting blood samples were collected to test the lipid profile. Information regarding fruit consumption and other characteristics was collected by questionnaires. Logistic regression models adjusting for confounding covariates were applied to calculate the odds ratio (OR) and 95% confidence interval (95% CI). Participants who consumed fruits for 6–7 days per week had lower risks of high triglycerides (OR: 0.66, 95% CI: 0.58–0.75), dyslipidemia (OR: 0.77, 95% CI: 0.68–0.86), and hyperlipidemia (OR: 0.72, 95% CI: 0.63–0.81), compared to fruit consumption of 0–2 days per week. Risks of high triglycerides, dyslipidemia and hyperlipidemia of those who consumed fruits for 0.75–1.5 servings each day also decreased, compared to the insufficient fruit intake. The combined effects of high frequency and moderate daily intake of fruit on lipid disorders did not change essentially. The associations were more evident in girls, younger children and those whose families had higher educational levels. Moderate fruit consumption was associated with lower odds of lipid disorders, predominantly in girls, younger participants, and those came from higher-educated families. These findings supported the health effect of moderate fruit intake frequently to improve the childhood lipid profiles.

## 1. Introduction

Dyslipidemia, characterized by abnormal blood lipids and lipoprotein levels, is a recognized risk factor for cardiovascular disease [1]. Previous studies have observed a high prevalence of lipid disorders during both childhood and adolescence in China [2,3]. Observed from the 10-year trend of serum lipids, the pooled prevalence of dyslipidemia in Chinese children and adolescents was 28.9% in 2014 [2], while in 2015, it increased to 31.6% in a nationally representative pediatric population with 129,426 participants [4]. It showed a younger-age trend during recent decades, and could be progressive into adulthood. Growing evidence indicated that, in children and adolescents, higher concentrations of low density lipoprotein-cholesterol (LDL-C), as well as lower concentrations of high density lipoprotein-cholesterol (HDL-C) were associated with higher risk of atherosclerosis in later life [5]. As such, screening children and adolescents for lipid profiles, which usually included the measurements of serum concentrations of total cholesterol (TC), triglycerides (TG), HDL-C and LDL-C, might have the potential to identify the affected subjects, reduce the burdens of long-term cholesterol through intervention, and postpone or prevent cardiovascular events during adulthood. The challenge was to maintain appropriate lipid levels at the right time, most commonly by early behavioral and lifestyle interventions for high-risk children and adolescents.

Previous work revealed that the nutritional and diet factors were important determinants for the development of childhood dyslipidemia [1,6], and therefore could be important targets for prevention strategies. Among the multiple diet factors, vegetables and fruits are important sources of a healthy diet. Notably, low fruit consumption is considered to be the fifth leading contributor to the global disease burden [7]. Growing observational evidence suggested that the fruit consumption might parallel the decrease in the risks of obesity, diabetes mellitus, and cardiovascular events in both childhood and adulthood [8,9]. Pan and his colleagues found a significant inverse association between healthy eating index score of fruits and the risk of metabolic syndrome (MS) among US adolescents, suggesting that a fruit-rich diet could exert beneficial effects in prevention of MS [10]. However, inadequate fruit consumption of children and adolescents worldwide despite the generally higher preference for consumption of fruits than vegetables [11], interventions to encourage fruit consumption during childhood and adolescence might therefore be an effective strategy in reducing disease burden.

So far, evidence for the effects of fruit consumption on lipid disorders had been limited to date. In 2013, a randomized controlled trial demonstrated no significant effect between increased fruits and vegetables consumption and healthier blood levels of TC [12]. In contrast, a cross-sectional study in Macao, China indicated that students who consumed less fruits suffered from a higher rate of low HDL-C level and elevated TG [13]. Besides, a fruits- and vegetables-rich diet was associated with a healthier metabolic profile, reflected by low concentrations of TC and LDL-C [14]. However, these studies only determined the consumption frequency of fruits, such as consuming fruits every day, not focused on the specific frequencies or average daily intake amount. Currently, the dietary guidelines for fruit intake were mostly based on adults, with the recommended daily fruit intake of 400 g [15], while there was no consensus regarding daily fruit consumption for children and adolescents [16], especially aimed at reducing adverse lipid profiles. Different from children and adolescents, adults could accumulate more complex environmental effects, while children might be more sugar-sensitive than adults. Guidelines based on findings in adults may therefore lead to ambiguity about fruit recommendations for pediatric cardiovascular and lipid health. In addition, fruit also contains a large amount of fructose, the accumulation of which is detrimental. Whether a causal link exists between natural sources of fructose present in fruits and the development of lipid disorders continues to be contested [17]. For these reasons, exploring the exact frequency and amount of fruit for children and adolescents to prevent lipid disorders was essential.

Given the role that fruit consumption played in pediatric health, we hereby investigated the relationship between fruit consumption and lipid profiles among children and adolescents aged 5–19 years, based on data of a large cross-sectional survey, which was conducted in seven provinces in China. The aim of the present study was to provide evidence on a satisfactory frequency and daily amount of fruit consumption to prevent deleterious lipid events for children and adolescents.

## 2. Materials and Methods

### 2.1. Study Population

Data in this study came from the baseline of a multi-centered, cluster-controlled trial, aiming to prevent obesity in children and adolescents from seven provinces or cities of China (Hunan, Ningxia, Hunan, Chongqing, Liaoning, Shanghai, and Guangzhou; registration number: NCT02343588). The full trial protocol, including sampling procedure and measurements has been published previously in detail [18]. In brief, a multi-stage cluster random sampling method was used to determine the original study population. Firstly, several regions were randomly chosen from each province. Approximately 12–16 schools covering primary schools, junior high schools, and middle high schools were then randomly chosen from each region. In each school, two classes were randomly selected from each grade. All the students and their parents from selected classes were invited for participation. Among the original surveyed population of 16,637 participants aged 5–19 years whose physical examination and blood samples were available, 1862 participants were excluded from the present analysis because of missing information on fruit consumption, making the final sample size of 14,755. The study has been approved by the ethical committee of the Peking University (number: IRB0000105213034). Written informed consent was obtained from both students and their parents or legal guardian in both waves.

### 2.2. Questionnaire

The children’s questionnaire was performed in order to collect basic information and lifestyle behaviors. Besides, the parental self-administrated questionnaire included information about residence area, monthly household income, parental body mass index (BMI), and parental educational attainment. Both parental and children’s questionnaires of children grade 1–3 were reported by parents. Children above the fourth grade would fill in children’s questionnaire by themselves instructed by the class teacher.

Data regarding consumption of fruits and other eating behaviors were collected. As previously published [18,19,20,21], the frequency (days) and amount (serving per day) of dietary behaviors, including the total consumption of fruits, vegetables, meat and sugar-sweetened beverages (SSBs) over the past 7 days, were investigated. Participants were asked “How many days, over the past 7 days, have you eaten fruit/vegetables/meat or drunk SSB? How many servings in one day?” [18,19,20,21]. Previous studies determined the consumption frequency of fruits as consuming fruits every day [13], in order to avoid extreme few samples in each group, we categorized the frequency of those dietary behaviors into three groups of “0–2 days/week”, “3–5 days/week”, and “6–7 days/week”. To better understand the intake of fruit/vegetable, one serving was defined as the size of an ordinary adult’s closed fist (Appendix A) and roughly equaled a medium-sized apple or orange (≈200 g) [22], which has been described in detail elsewhere [21]. As set in the questionnaire, SSB included Coca-Cola, Sprite, orange juice, Nutrition Express, and Red Bull [20]. One serving of SSB was determined as a canned beverage (approximately 250 mL), while one portion of meat equaled the size of an adult’s palm (approximately 100 g) [23]. The daily dietary intake was calculated as: average daily intake  =  (days of consumption × servings in those days)/7.

According to the Dietary Guidelines for Chinese School-age Children 2016 [24], inadequate daily fruit intake was defined as eating fruits less than 150 g each day, and the recommended daily intake was approximately 250–350 g for the pediatric population. The amount of daily fruit intake was therefore categorized into groups of “<0.75 servings/day (approximately <150 g)”, “0.75–1.5 servings/day (≥0.75 and <1.5 servings/day) (approximately 150–300 g)”, and “≥1.5 servings/day (approximately ≥300 g)”.

Information about the child’s physical activity was collected with the International Physical Activity Questionnaire-Short Form (IPAQ-SF) [25], which is more suitable for population surveillance and large-scale studies [26]. Instructed by trained project members, all recruited participants were asked to report their frequency (days) and duration (hours and minutes) of moderate to vigorous-intensity physical activities (MVPA) over the past 7 days, and the average time for MVPA per day was calculated as: average daily time  =  (days of physical activity × duration in those days)/7.

Parents were asked to report their height (cm) and weight (kg), while body mass index (BMI) was calculated as the weight (kg) divided by the square of the height (m^2^). According to the criteria established by the Working Group on Obesity in China (WGOC) for Chinese adults [27], BMI cut-offs of 24 and 28 kg/m^2^ were applied to categorize each parent into the categories of normal, overweight and obesity. Parental educational attainment was surveyed and grouped into “primary school or below”, “secondary or equivalent” and “junior college or above”. In addition, the residence area was divided into “rural” and “urban”, and monthly household income was defined with the sum of monthly income (in CNY) of all household members and then classified into <5000, or ≥5000 CNY.

### 2.3. Anthropometric Measurements

Anthropometric measurements were conducted by trained project members according to standardized procedure, and the measuring instruments were similar at all study sites. Children were required to stand straight in light clothing and without shoes for the measurements. Height was measured using the portable stadiometer with 0.1 cm precision, weight was measured to the nearest 0.1 kg by a Lever type weight scale. Every indicator was measured twice, and the average level of the two measurements was calculated for final analyses. BMI was calculated as body weight (kg) divided by height (m) squared.

### 2.4. Blood Sample Collection and Detection

Venous blood sample was obtained in the morning after an overnight (at least 8 h) fasting. Children were asked to rest for at least 10 min before blood sample collection. Blood specimens were then transported in a chilled insulated container immediately, and then centrifuged at 3000 rpm for 10 min and then frozen at −80 °C. All plasma samples were transported by air in dry ice to the laboratory in Beijing, where the samples were stored at −80 °C before laboratory detections. All the biochemical analyses were conducted at a biomedical analyses company accredited by Peking University [18]. Lipid profiles were measured with an autoanalyzer (TBA-120FR, Toshiba, Tokyo, Japan), with TC and TG assayed by enzymatic method, while LDL-C and HDL-C measured by clearance method.

### 2.5. Definition of Abnormal Lipid Profile and Overweight/Obesity

Following the 2011 Expert Panel on Integrated Guidelines for Cardiovascular Health and Risk Reduction in Children and Adolescents [28], high TG was defined as TG ≥ 1.13 mmol/L for children aged 9 years or younger, and TG ≥ 1.47 mmol/L for adolescents aged 10 years or older. High TC was considered as TC ≥ 5.18 mmol/L, high LDL-C was regarded as LDL-C ≥ 3.37 mmol/L, and low HDL-C referred to HDL-C ≤ 1.04 mmol/L. Because non-HDL-C levels showed better prediction of persistent dyslipidemia, the prevalence of high non-HDL-C levels was also presented, as was calculated as the difference between TC level and the HDL-C level. High non-HDL-C were determined as non-HDL-C ≥ 3.76 mmol/L [28]. A participant with one or more abnormal lipid levels (high LDC-C, low HDL-C, high TG, or high TC) was defined as having dyslipidemia. In addition, hyperlipidemia was defined with the presence of high TG or high TC levels [28].

In accordance with the guideline of the Working Group on Obesity in China, participants with age- and sex-specific BMI < 85th percentile were considered as non-overweight/obese, and those with ≥85th percentile BMI were considered as overweight/obese [29].

### 2.6. Statistical Analysis

Data were expressed as mean ± SD for continuous variables and number (%) for categorical variables. We used Bonferroni multiple comparison methods to examine the differences in prevalence of adverse lipid profiles between groups. The significance of Bonferroni formula was set at α = 0.017. To understand whether some demographics, dietary or lifestyle factors could modify the associations, two separate logistic regression models were applied to estimate the odds ratio (OR) and 95% confidence interval (95% CI) between fruit consumption and lipid profile. In model 1, age and residence area were adjusted. In model 2, several additional confounding factors, including sex (boy, girl), BMI values, ethnicity (Han, Hui, Tibetan, Mongolian and Other), monthly household income (<5000, or ≥5000 CNY), parental weight status (normal, overweight or obesity), parental educational attainment (primary school or below, secondary or equivalent, junior college or above), vegetable consumption, sugar-sweetened beverages consumption, meat consumption and physical activity were included. Additionally, previous studies had shown that lipid levels were dependent on age and sexual maturation [30], and could be influenced by the difference of socioeconomic support mainly determined by the degree of their parents’ education [31]. Besides, BMI values were closely related with lipid health. Therefore, stratified analyses were conducted according to sex, age, parental educational attainment and BMI values.

All analyses were performed using SAS software (version 9.4; SAS Institute, lnc., Cary, NC, USA). A two-sided *p* value < 0.05 was considered statistically significant.

## 3. Results

### 3.1. General Characteristics

A total of 7420 boys and 7335 girls were included in the final analysis. Table 1 showed the characteristics of the study population. The mean age was 11.15 ± 3.29 years old and their average BMI value was 18.74 ± 3.85 kg/m^2^. The majority of the study population was Han ethnicity (92.56%), and most of their parents belonged to the normal weight status (paternal: 56.88%; maternal: 79.49%). In addition, 35.33% of the children’s fathers received junior college or above education, 58.33% for secondary or equivalent and 6.34% for primary school or below. A similar trend was observed for maternal educational attainment. It was worth noting that most of them had a monthly household income of less than 5000 yuan. Besides, the average concentrations of TC, TG, LDL-C, HDL-C and non-HDL-C were 3.89 ± 0.88, 1.09 ± 0.77, 2.01 ± 0.69, 1.90 ± 1.35 and 1.99 ± 1.56 mmol/L, respectively.

Furthermore, girls tended to consume fruits and vegetables more frequently per week (*p* < 0.05), while boys were more likely to consume sugar-sweetened beverages and meat (*p* < 0.01). As for the average daily consumption, intakes of fruits (*p* = 0.567) and vegetables (*p* = 0.148) were not significantly different among sex-specific groups, but boys consumed more sugar-sweetened beverages and meat daily (both *p* < 0.01). Of note, boys were prone to exercise more frequently and for longer duration (both *p* < 0.01).

### 3.2. The Prevalence of Abnormal Lipid Profile

The prevalence of abnormal lipid profile was present in Appendix A. Compared with participants consuming fruits for only 0–2 days/week, the prevalence of high TG, low HDL-C, dyslipidemia, and hyperlipidemia tended to be low among those who consumed fruits more frequently. Notably, similar lower prevalence for these lipid disorders was observed in the group of average daily fruit intake of 0.75–1.5 servings, compared to those with daily fruit consumption of ≥1.5 servings. Similar trends of prevalence of adverse lipid profiles were found for boys and girls, respectively.

### 3.3. Association between Fruit Consumption and Lipid Profile

The associations between fruit consumption and the odds of abnormal lipid profile were present in Table 2, high frequency of fruit intake per week was associated with lower possibility of unfavorable lipid profiles. In the basically adjusted model 1, the ORs (95% CI) of frequent fruit consumption (6–7 days/week) were 0.74 (0.66–0.83), 0.83 (0.75–0.92) and 0.79 (0.70–0.88) for the risk of high TG, dyslipidemia and hyperlipidemia, respectively. After controlling for potential covariates included in model 2, the results did not change essentially.

In addition, adjusting for factors in model 2, participants who consumed fruits moderately for 0.75–1.5 servings each day were estimated to have 0.87 times (OR: 0.87, 95% CI: 0.78–0.97), 0.88 times (OR: 0.88, 95% CI: 0.81–0.97) and 0.88 times (OR: 0.88, 95% CI: 0.80–0.97) lower odds of high TG, dyslipidemia and hyperlipidemia compared with their counterparts with insufficient fruit consumption each time. There were no statistically significant associations between fruit consumption over 1.5 servings/day and risks of high TC, high LDL-C, low HDL-C and high non-HDL-C.

Apart from this, consuming fruits for 6–7 days/week combined with moderate intake of 0.75–1.5 servings each day could improve adverse lipid profiles (Table 3). Children and adolescents who consumed fruits moderately of 0.75–1.5 servings/day for 6–7 days/week had lower risks of high TG, low HDL-C, dyslipidemia and hyperlipidemia (all *p* < 0.05).

### 3.4. Stratified Analyses

The results of stratified analyses were similar with the main results (Figure 1 and Figure 2 and Appendix A). The more frequent of fruit intake was associated with lower odds of high TG, dyslipidemia and hyperlipidemia both in boys and girls, while moderately consuming fruits each day could only lower the likelihood of lipid disorders among girls (Figure 1). Notably, compared to the reference group, significantly lower likelihoods of high TG and hyperlipidemia were observed among all participants aged 5–19 years old who consumed fruits more frequently, but only 5–14-year-old children and adolescents had a lower risk of dyslipidemia (*p* < 0.01). However, when considering consuming fruits moderately of 0.75–1.5 servings per day, only 5-to-9-year-old children were estimated to have 0.80 times (OR: 0.80, 95% CI: 0.66–0.96), 0.79 times (OR: 0.79, 95% CI: 0.68–0.92) and 0.77 times (OR: 0.77, 95% CI: 0.66–0.91) lower odds of high TG, dyslipidemia and hyperlipidemia, respectively (Figure 2).

When the study sample was restricted according to parental educational attainment, the main associations did not change essentially, but were more significantly pronounced among participants from families with higher parental education levels, compared with those from lower educational families (Appendix A). Notably, the results of the two subgroups presented similar according to BMI values (Appendix A).

## 4. Discussion

Based on the national representative sample, our findings suggested significant inverse associations between high frequency of fruit consumption per week and moderate daily fruit intake with the odds of adverse lipid profiles. The associations were more pronounced among girls, younger participants and those whose families had a higher educational background, when presented similar by BMI subgroups. These findings supported the health effect of moderate fruit intake frequently to improve the childhood lipid profile.

Our findings were in accordance with previous studies in direction. A case-control study in Jordanians revealed that consumption of banana could reduce the odds of cardiovascular events to about 44% and 62% during adulthood when consuming 1–2 and 3–6 servings per week, respectively [32]. Growing reviews and meta-analyses found consistent associations between fruit consumption and cardiovascular events, diabetes, cancer, and other chronic diseases [33,34]. Among a pediatric population, a diet rich in dietary fiber from fruits was beneficial to a healthy cardiovascular profile, regardless of European children’s weight [35]. Since there was evidence supporting that eating behaviors of children and adolescents were likely to persist into adulthood [36], encouraging healthy eating habits among school-aged children might therefore represent an effective primary prevention strategy for reducing the risk of chronic diseases. As recommended by ISPAD Clinical Practice Consensus Guidelines 2018 [37], intake of a variety of fruit should be encouraged among children and adolescents with diabetes, which could be particularly useful in helping to reduce lipid levels.

Although WHO recommends 5 servings (400 g) of fruits and vegetables per day for the overall population [15], among the pediatric population, studies rarely applied the criteria of 5 servings per day since only few children and adolescents met this recommendation [16,38], thus it was unlikely that any significant association between the recommended levels and lipid health would be detected. Therefore, the absence of a specific frequency and daily amount of fruit consumption for preventing pediatrics’ lipid disorders was an important limitation. In the present study, daily fruit consumption of 0.75–1.5 servings (approximately 150–300 g) for 6–7 days/week was crucial for Chinese pediatrics’ lipid health. We speculated that the continued increase in fruit intake would not always promote lipid health, which might be explained by the following reasons: Fructose is a kind of sugar found in fruits, the excess accumulation of which might be positively associated with the level of plasma TG and atherogenic indices [39]. Besides, it could disrupt the utilization of dietary copper, while copper deficiency could produce high TC and high TG [40]. Since there was no consensus regarding the method of assessing fruit consumption in children and adolescents [41], additional studies with high methodological quality are needed to develop informed guidelines for fruit consumption among pediatric population and to better promote lipid health.

However, the exact mechanisms by which increased fruit consumption reduced abnormal lipid profile were not fully understood. As for these inverse associations, antioxidant compounds and polyphenols found in fruit (e.g., vitamin C, carotenoids, and flavonoids) could prevent the oxidation of cholesterol in the arteries [42]. In addition, those compounds could reduce systemic inflammation through cellular signaling processes, therefore preventing atherosclerosis and cardiovascular disease [43]. Apart from this, several components of fruit have cholesterol lowering properties, particularly dietary fiber, as well as high amounts of water and low amounts of saturated fat, which are associated with reduced energy density, hunger control, and satiety [44]. Taken together, the evidence accumulated so far does support that fruit consumption may reduce the odds of lipid disorders.

Subgroup analyses suggested that the associations between moderate fruit consumption each day and the odds of lipid disorders were more pronounced in girls. A previous study conducted in Korea also emphasized the sex differences, and concluded that low fruit intakes were significantly associated with overweight and obesity among adolescent girls but not in boys [45]. To be noted, we found more frequent of fruit/vegetable consumption and less intake of SSBs among girls; in this context, different behavioral factors may influence the associations in girls and boys. Another potential behavioral explanation was that higher fruit consumption might reflect their health consciousness, consequently, a lower intake of sugar- and fat-rich products, especially in girls.

An interesting age effect was also present; this possibility indicated the potential implications of different lifestyle behaviors in school and preschool children. Our present study indicated these associations were more evident in those aged 5–14 years old. One of the reasons might be that younger children lived healthier lifestyles, while older children and adolescents were under heavier academic pressure, which often led them to overconsume SSBs and meat, as well as to reduce their outdoor activities (Appendix A). Notably, few studies investigated the association between fruit intakes and lipid profile among different age groups. Therefore we could not discuss an age-specific association between low fruit intakes and adverse lipid profile based on the previous findings.

Targeting the family environment for the promotion of lipid health among children and adolescents is important. Our findings were in agreement with previous surveys which concluded that family education could potentially modify the relationships between dietary behaviors and risks of chronic diseases [30]. It was necessary to influence the fruit consumption preferences of their parents, in order to increase the fruit intake of children [46]. Higher parental educational background may be related to higher household income and thus greater availability of healthy foods, increased nutrition knowledge and higher motivation to follow a healthy lifestyle. In addition, we proposed that health and education programs should consider these findings and be implemented widely to make the public aware of the importance of moderate fruit consumption for the pediatric population. In addition, the similarity of subgroup analysis by BMI showed that the relationship between fruit consumption and lipid profiles was independent of BMI values, to a certain extent. Children and adolescents should consume fruits moderately each day to maintain lipid health, regardless of their weight status.

Our study had strengths of the national representative sample from seven provinces in China, and we focused on 5–19-year-old children and adolescents. However, there were also limitations that should be taken into account when interpreting the findings. Firstly, since 92.56% of the study population was of Han ethnicity, our results may not be applicable to other ethnic groups. Secondly, we only relied on self-reported dietary intake, which could lead to a certain degree of recall bias. However, in this process, we carried out strict quality control to ensure the reliability. In addition, the dietary data recall of 7 days might not represent long-term dietary behaviors; thus, in future studies, a pilot study using food tracking, such as photo tracking, could be used to accurately assess the frequency and amount of fruit and other dietary intake in individuals. Thirdly, this was a cross-sectional survey and thus could not generate casual relationships of fruit consumption and lipid profile. A randomized controlled trial was desired to confirm our results. Fourth, measuring fruit consumption was complex, with interactions of different fruit compounds, cultural and socioeconomic conditions [47]. Since it was difficult to collect accurate and detailed information on intake of various types of food in a large population survey, we could not identify the specific type of fruits consumed by the participants, and different fruits might have various influences on lipid profiles. However, as for food and diet information, both parent and child questionnaire of children grade 1–3 were reported by parents. In addition, trained project members interpreted all the questionnaires in detail. Appropriate guidance would be given by these project members as effectively as possible. The questionnaires would be rechecked by 3% within one week for the same participants [18]. Therefore, the quality of diet information was guaranteed largely. Besides, other important dietary factors, such as macronutrient and energy intakes, were not available in the present study. In order to minimize possible influence of these factors, we included vegetable, meat and SSBs consumption in the adjusted model. Fifth, the IPAQ-SF questionnaire used in this study was not completely suitable for the assessment of physical activity of children and adolescents. However, due to the open-ended questions of IPAQ-SF, and appropriate guidance would be given by project members, to a certain extent, the IPAQ-SF could reflect the levels of physical activity of children and adolescents. Further studies with more information regarding confounding factors and eating behaviors were desired in the future.

## 5. Conclusions

Our findings confirmed that a diet rich in fruit could be effective for improving lipid health. Given the reason that the basis of dietary intake recommendations is related with its health implications, moderate fruit consumption of approximately 150–300 g each day for 6–7 days/week could be beneficial to lipid health for children and adolescents. Monitoring dietary habits early in childhood and adolescence might have a positive impact on lipid health and overall quality of life. For the pediatric population, indicators of family circumstances should also be applied to identify target groups for interventions aimed at promoting eating habits and lipid health.

## Figures and Tables

**Figure 1 nutrients-14-00063-f001:**
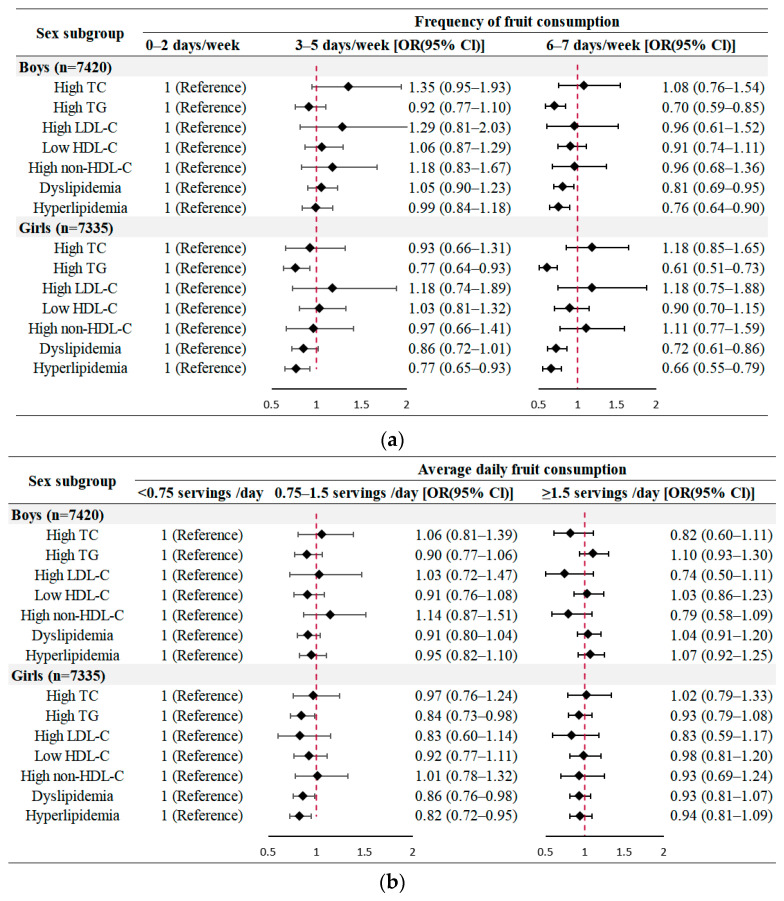
Sex-specific analysis of the associations between fruit consumption and lipid profile (adjusted for age, BMI values, residence area, ethnicity, incomes, parental educational attainment, parental weight, vegetable consumption, sugar-sweetened beverages consumption, meat consumption and physical activity) (**a**) Frequency of fruit consumption; (**b**) Average daily fruit consumption. TC, total cholesterol; TG, triglycerides; LDL-C, low density lipoprotein-cholesterol; HDL-C, high density lipoprotein-cholesterol; diamond symbol, point estimates; red dashed line, invalid line.

**Figure 2 nutrients-14-00063-f002:**
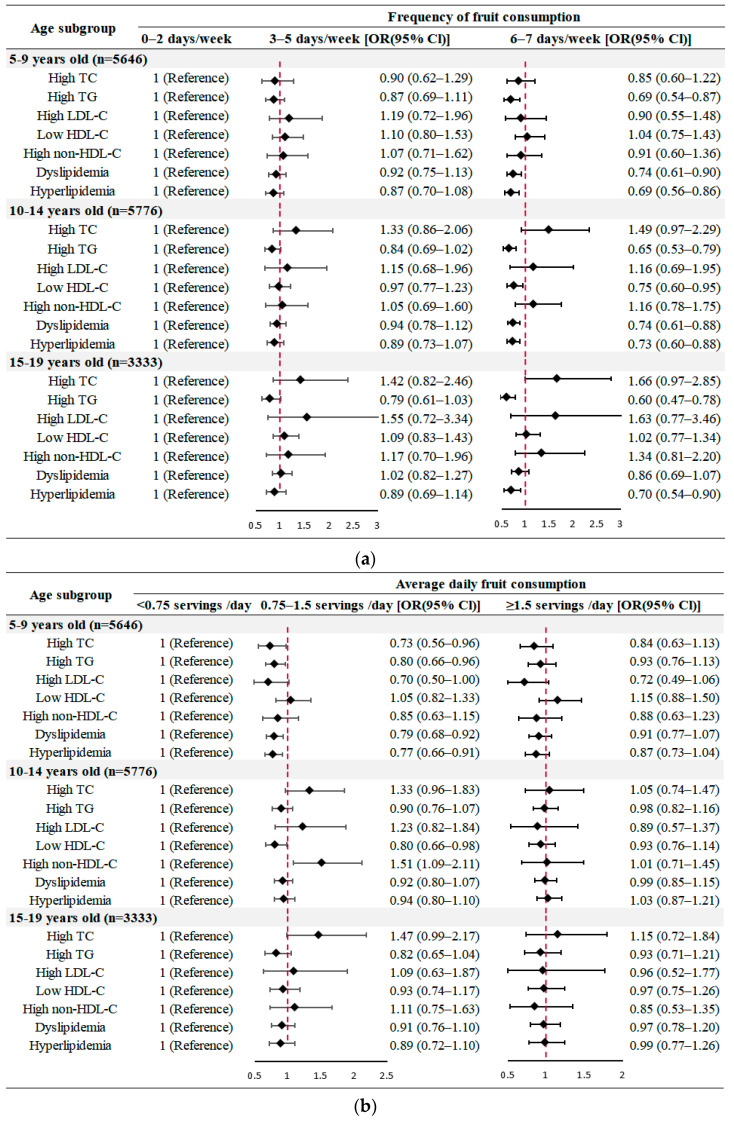
Age-specific analysis of the associations between fruit consumption and lipid profile (adjusted for sex, BMI values, residence area, ethnicity, incomes, parental educational attainment, parental weight, vegetable consumption, sugar-sweetened beverages consumption, meat consumption and physical activity) (**a**) Frequency of fruit consumption; (**b**) Average daily fruit consumption. TC, total cholesterol; TG, triglycerides; LDL-C, low density lipoprotein-cholesterol; HDL-C, high density lipoprotein-cholesterol; diamond symbol, point estimates; red dashed lines, invalid line.

**Table 1 nutrients-14-00063-t001:** Baseline characteristic of included population.

Characteristics	Total Population (*n* = 14,755)	Boys (*n* = 7420)	Girls (*n* = 7335)	*p*-Value
Age, year	11.15 ± 3.29	11.13 ± 3.25	11.16 ± 3.33	0.533
Weight, kg	42.27 ± 15.91	44.09 ± 17.46	40.45 ± 13.94	<0.01
Height, m	1.48 ± 0.17	1.49 ± 0.18	1.46 ± 0.15	<0.01
BMI, kg/m^2^	18.74 ± 3.85	19.04 ± 4.04	18.45 ± 3.63	<0.01
Residence area (*n*, %)				0.701
Urban	7860 (53.27%)	3941 (53.11%)	3919 (53.43%)	
Rural	6895 (46.73%)	3479 (46.89%)	3416 (46.57%)	
Ethnicity (*n*, %)				0.066
Han	13,657 (92.56%)	6876 (92.67%)	6781 (92.45%)	
Hui	528 (3.58%)	248 (3.34%)	280 (3.82%)	
Tibetan	33 (0.22%)	14 (0.19%)	19 (0.26%)	
Mongolian	187 (1.27%)	110 (1.48%)	77 (1.05%)	
Other	350 (2.37%)	172 (2.32%)	178 (2.43%)	
Paternal weight status, *n* (%)				0.451
Normal	8393 (56.88%)	4252 (57.30%)	4141 (56.46%)	
Overweight	4834 (32.76%)	2395 (32.28%)	2439 (33.25%)	
Obesity	1528 (10.36%)	773 (10.42%)	755 (10.29%)	
Maternal weight status, *n* (%)				0.002
Normal	11,729 (79.49%)	5983 (80.63%)	5746 (78.34%)	
Overweight	2462 (16.69%)	1176 (15.85%)	1286 (17.53%)	
Obesity	564 (3.82%)	261 (3.52%)	303 (4.13%)	
Paternal educational attainment, *n* (%)				0.229
Primary school or below	936 (6.34%)	451 (6.08%)	485 (6.61%)	
Secondary or equivalent	8606 (58.33%)	4371 (58.91%)	4235 (57.74%)	
Junior college or above	5213 (35.33%)	2598 (35.01%)	2615 (35.65%)	
Maternal educational attainment, *n* (%)				0.385
Primary school or below	1277 (8.65%)	619 (8.34%)	658 (8.97%)	
Secondary or equivalent	8429 (57.13%)	4246 (57.22%)	4183 (57.03%)	
Junior college or above	5049 (34.22%)	2555 (34.43%)	2494 (34.00%)	
Monthly household income, *n* (%)				0.220
<5000 yuan	12,402 (84.05%)	6264 (84.42%)	6138 (83.68%)	
≥5000 yuan	2353 (15.95%)	1156 (15.58%)	1197 (16.32%)	
Frequency of consumption (days per week)				
Fruit	4.93 ± 2.11	4.75 ± 2.19	5.11 ± 2.02	<0.01
Vegetables	6.02 ± 1.82	5.99 ± 1.85	6.06 ± 1.79	0.011
Sugar-sweetened beverages	1.63 ± 1.78	1.84 ± 1.91	1.41 ± 1.62	<0.01
Meat	5.01 ± 2.24	5.21 ± 2.18	4.80 ± 2.28	<0.01
Average daily consumption (servings per day)				
Fruit	1.33 ± 1.14	1.32 ± 1.19	1.33 ± 1.08	0.567
Vegetables	1.87 ± 1.50	1.89 ± 1.53	1.85 ± 1.46	0.148
Sugar-sweetened beverages	0.46 ± 0.84	0.56 ± 0.96	0.36 ± 0.68	<0.01
Meat	1.24 ± 1.30	1.39 ± 1.42	1.09 ± 1.15	<0.01
Frequency of physical activity, days/week	3.35 ± 2.53	3.47 ± 2.55	3.23 ± 2.50	<0.01
Average daily physical activity, hours and minutes	0.37 ± 0.82	0.42 ± 0.89	0.32 ± 0.74	<0.01
TC, mmol/L	3.89 ± 0.88	3.83 ± 0.87	3.94 ± 0.89	<0.01
TG, mmol/L	1.09 ± 0.77	1.04 ± 0.74	1.14 ± 0.79	<0.01
LDL-C, mmol/L	2.01 ± 0.69	1.98 ± 0.67	2.03 ± 0.70	<0.01
HDL-C, mmol/L	1.90 ± 1.35	1.90 ± 1.38	1.90 ± 1.33	0.993
Non-HDL-C, mmol/L	1.99 ± 1.56	1.94 ± 1.58	2.04 ± 1.54	<0.01

Abbreviation: BMI, body mass index; TC, total cholesterol; TG, triglycerides; LDL-C, low density lipoprotein-cholesterol; HDL-C, high density lipoprotein-cholesterol.

**Table 2 nutrients-14-00063-t002:** Multivariate odds ratios (OR) and 95% confidence intervals (CI) of fruit consumption and lipid profile (*n* = 14,755).

Lipid Profile	Frequency of Fruit Consumption			Average Daily Fruit Consumption		
	0–2 Days/Week	3–5 Days/Week	6–7 Days/Week	<0.75 Servings/Day	0.75–1.5 Servings/Day	≥1.5 Servings/Day
Model 1 ^1^						
High TC	1 (Reference)	1.09 (0.86–1.38)	1.17 (0.93–1.46)	1 (Reference)	1.01 (0.85–1.20)	0.93 (0.78–1.12)
High TG	1 (Reference)	0.88 (0.79–0.99) *	0.74 (0.66–0.83) **	1 (Reference)	0.92 (0.83–1.01)	1.09 (0.99–1.21)
High LDL-C	1 (Reference)	1.14 (0.84–1.55)	1.13 (0.84–1.53)	1 (Reference)	0.93 (0.75–1.17)	0.86 (0.67–1.09)
Low HDL-C	1 (Reference)	0.96 (0.84–1.11)	0.88 (0.76–1.01)	1 (Reference)	0.94 (0.83–1.05)	1.06 (0.94–1.19)
High non-HDL-C	1 (Reference)	0.99 (0.78–1.26)	1.05 (0.83–1.32)	1 (Reference)	1.07 (0.89–1.28)	0.90 (0.74–1.10)
Dyslipidemia	1 (Reference)	0.97 (0.87–1.07)	0.83 (0.75–0.92) **	1 (Reference)	0.93 (0.86–1.01)	1.05 (0.96–1.15)
Hyperlipidemia	1 (Reference)	0.92 (0.82–1.03)	0.79 (0.70–0.88) **	1 (Reference)	0.92 (0.84–1.01)	1.07 (0.97–1.17)
Model 2 ^2^						
High TC	1 (Reference)	1.13 (0.88–1.45)	1.18 (0.92–1.50)	1 (Reference)	1.01 (0.85–1.22)	0.94 (0.77–1.14)
High TG	1 (Reference)	0.84 (0.74–0.96) *	0.66 (0.58–0.75) **	1 (Reference)	0.87 (0.78–0.97) **	1.00 (0.90–1.13)
High LDL-C	1 (Reference)	1.23 (0.89–1.71)	1.09 (0.79–1.51)	1 (Reference)	0.92 (0.73–1.17)	0.80 (0.62–1.03)
Low HDL-C	1 (Reference)	1.02 (0.88–1.19)	0.88 (0.76–1.03)	1 (Reference)	0.91 (0.80–1.03)	1.00 (0.87–1.14)
High non-HDL-C	1 (Reference)	1.07 (0.83–1.38)	1.05 (0.82–1.35)	1 (Reference)	1.08 (0.89–1.31)	0.87 (0.70–1.07)
Dyslipidemia	1 (Reference)	0.95 (0.85–1.06)	0.77 (0.68–0.86) **	1 (Reference)	0.88 (0.81–0.97) **	0.98 (0.89–1.08)
Hyperlipidemia	1 (Reference)	0.88 (0.78–1.00)	0.72 (0.63–0.81) **	1 (Reference)	0.88 (0.80–0.97) **	1.00 (0.90–1.11)

^1^ Model 1: adjusted for age and residence area. ^2^ Model 2: additionally adjusted for sex, BMI values, ethnicity, incomes, parental educational attainment, parental weight, vegetable consumption, sugar-sweetened beverages consumption, meat consumption and physical activity. TC, total cholesterol; TG, triglycerides; LDL-C, low density lipoprotein-cholesterol; HDL-C, high density lipoprotein-cholesterol. * *p* < 0.05, ** *p* < 0.01.

**Table 3 nutrients-14-00063-t003:** Combined effects of frequencies and daily intake of fruit consumption on adverse lipid profile (*n* = 14,755).

Lipid Profile	Unselected Population	6–7 Days/Week (*n* = 6829)		
		<0.75 Servings/Day (*n* = 61)	0.75–1.5 Servings/Day (*n* = 3036)	≥1.5 Servings/Day (*n* = 3732)
High TC	1 (Reference)	0.48 (0.11–2.04)	1.10 (0.92–1.32)	1.01 (0.85–1.21)
High TG	1 (Reference)	0.78 (0.36–1.68)	0.68 (0.61–0.77) **	0.94 (0.85–1.05)
High LDL-C	1 (Reference)	0.71 (0.15–3.23)	1.03 (0.81–1.31)	0.90 (0.71–1.13)
Low HDL-C	1 (Reference)	0.86 (0.35–2.09)	0.83 (0.73–0.96) *	0.97 (0.86–1.09)
High non-HDL-C	1 (Reference)	0.75 (0.21–2.73)	1.12 (0.92–1.36)	0.91 (0.75–1.09)
Dyslipidemia	1 (Reference)	0.69 (0.36–1.33)	0.76 (0.69–0.83) **	0.95 (0.87–1.04)
Hyperlipidemia	1 (Reference)	0.68 (0.33–1.41)	0.72 (0.65–0.81) **	0.96 (0.88–1.06)

Adjusted in Model 2: including age, residence area, sex, BMI values, ethnicity, incomes, parental educational attainment, parental weight, vegetable consumption, sugar-sweetened beverages consumption, meat consumption and physical activity. TC, total cholesterol; TG, triglycerides; LDL-C, low density lipoprotein-cholesterol; HDL-C, high density lipoprotein-cholesterol. * *p* < 0.05, ** *p* < 0.01.

## Data Availability

The data supporting the conclusions of this article will be made available from the corresponding author upon request.

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
