# Peer review of "Association between Fruit Consumption and Lipid Profile among Children and Adolescents: A National Cross-Sectional Study in China"

_nutrients, 2021, doi:10.3390/nu14010063_

Round 1

Reviewer 1 Report

Overall, an interesting paper with a few points that still need to be addressed, before being accepted for publication. Minor spelling and grammatical errors identified throughout the text.

Introduction

Line 40: The study by Pan et al. 2016 (reference 3) was conducted in adults and not in children/ adolescents. Please revise.

Materials and methods

Lines 89-91: Please state the aim of the controlled trial you used baseline data from, in a clearer way. “Addressing the intervention of obesity” is a poor description. Potentially an intervention to reduce prevalence of obesity?

Lines 103-104: It is mentioned that informed consent was obtained both from students and their parents, however in your study you have included children as young as 5 years old, who are unable to provide informed consent. In studies involving children it is necessary to obtain informed consent from the parents or legal guardian and assent consent either orally or in writing from the child. Please clarify which was the case in this study, as this is a serious ethical issue.

Lines 119-120: one fruit/ vegetable serving is defined as 1 cup ≈200 g, however this sounds rather a lot for one serving. In addition, I could not find this information on reference 18 as quoted within text. Please clarify the reference for this measure.

Lines 133-135: The correct name of the questionnaire is IPAQ-SF (International Physical Activity Questionnaire - Short Form) and not IPAQ-S. As far as I know this questionnaire is recommended for use in people between 15-69 years of age. Do you think this is an appropriate tool for your cohort? In addition, you mention that it has been widely used in children and adolescents. Can you provide references? Preferably validation studies, mentioning its suitability for the age group included in this trial.

Lines 135-137: You mention that the students reported the frequency and duration of their physical activity. Was this the case even for the younger children of your cohort? I have concerns as to how accurately can a 5-year-old provide this information.

Results

On Table S1 we can see if there is a statistically significant difference between groups, however we cannot see where the difference lies. I suggest to run additional statistical analysis (in case it was not done) and improve clarity of data presented, by including information on statistical significance between groups.

Lines 235-241: Please clarify these are the results for model 2 and mention that this was not the case for model 1.

Table 3 legend: please mention that this refers to model 2.

Discussion

Please elaborate a bit more on the reasons why you think that beneficial effects on lipid profile were mainly detected for 0.75-1.5 servings of fruit/ day, but not for ≥1.5 servings/ day.

You mention that one of the strengths of you study is the large sample size from seven provinces in China, thus results "might be generalised to the general paediatric population". At another point on your manuscript, you mention that 92.56% of the study population was of Han ethnicity. Please provide evidence and explain how the results you found could be applicable to other ethnic groups.

Since nutritional data were self-reported, could you please elaborate a bit more on how definition of a serving size was communicated to participants? Previous research shows that there are large interindividual differences on the perception of portion sizes.

Reviewer 2 Report

The authors investigated the relationship between fruit consumption and the lipid profile among children and adolescents aged 5-19 using data from a large cross-sectional study. I have general doubts about the way of data interpretation and analysis. The respondents did not eat only fruit, so the analysis of the results could be carried out in terms of lifestyle (diet and physical activity - I know that the model was adjusted for other dietary components and PA), which is a determinant of health, including the lipid profile. Moreover, there is no comparison of the lipid profile with the BMI of children. I could not find information in the methodology of how the data on the weight and height of the children were collected. Did the questions about fruit consumption also include juices? To my knowledge, children have a greater problem with low vegetable consumption, so why there are fruits that children eat more often. This can be mentioned in the introduction.

In verses 74-75 you wrote: "... although there was no consensus on the daily consumption of fruit by children and adolescents" and in 127-129: "According to the 2016 Dietary Guidelines for Chinese School Children [21], inadequate daily intake fruit is defined as eating less than 150 grams per day, and the recommended daily intake is around 300 grams for the pediatric population ", and in conclusion, you repeated this recommendation. Please add an explanation of what is new from your study.

Author Response

Minor Revision

#Reviewer 2

The authors investigated the relationship between fruit consumption and the lipid profile among children and adolescents aged 5-19 using data from a large cross-sectional study. I have general doubts about the way of data interpretation and analysis. The respondents did not eat only fruit, so the analysis of the results could be carried out in terms of lifestyle (diet and physical activity - I know that the model was adjusted for other dietary components and PA), which is a determinant of health, including the lipid profile. Moreover, there is no comparison of the lipid profile with the BMI of children. I could not find information in the methodology of how the data on the weight and height of the children were collected.

Response: Thanks for your valuable suggestion. We acknowledged that the influencing factors among the studies on fruit consumption are complex, and it’s difficult to state which is a determinant of health. In the present study, we emphasized that only the associations between fruit consumption and adverse lipid profiles were observed, however we did not indicate that fruit consumption was a determinant of health.

We truly agree with your advice that BMI could influence the lipid profiles. Following the suggestion, we conducted main analyses which were additionally adjusted for BMI levels, while the main results did not change essentially. We have updated the results accordingly in all Tables and Figures.

In addition, following your suggestion, we did additional BMI-specific analyses in Figure S3 (a) and Figure S3 (b). The results showed that, the associations between fruit consumption and lipid profiles presented similar among BMI-specific analysis. The similarity of subgroup analysis by BMI showed that the relationship between fruit consumption and lipid profiles was independent of BMI values, to a certain extent. Children and adolescents should consume fruits moderately each day to maintain lipid health, in regardless of their weight status.

Accordingly, we have revised the Results and Discussions, and added the information in the methodology as follows. Based on your suggestion, the paper is now much improved:

2.3. Anthropometric Measurements

Anthropometric measurements were conducted by trained project members according to standardized procedure, and the measuring instruments were similar at all study sites. Children were required to stand straight in light clothing and without shoes for the measurements. Height was measured using the portable stadiometer with 0.1 cm precision, weight was measured to the nearest 0.1 kg by a Lever type weight scale. Every indicator was measured twice, and the average level of the two measurements were calculated for final analyses. BMI was calculated as body weight (kg) divided by height (m) squared. (line 162-170)

2.5. Definition of abnormal lipid profile and overweight/obesity

In accordance with the guideline of the Working Group on Obesity in China, participants with age- and sex-specific BMI <85th percentile were considered as non-overweight/obese, those with ≥85th percentile BMI were considered as overweight/obese[29]. (line 196-199)

2.6. Statistical analysis

Besides, BMI values were closely related with lipid health. Therefore, stratified analyses were conducted according to sex, age, parental educational attainment and BMI values. (line 216-218)

Results: stratified analyses

Notably, the results of the two subgroups presented similar according to BMI values (Figure S3). (line 292-293)

Discussion

The associations were more pronounced among girls, younger participants and those whose families had higher educational background, while presented similar by BMI subgroups. (line 305-307)

In addition, the similarity of subgroup analysis by BMI showed that the relationship between fruit consumption and lipid profiles was independent of BMI values, to a certain extent. Children and adolescents should consume fruits moderately each day to maintain lipid health, in regardless of their weight status. (line 383-386)

Reference in the manuscript:

[29] Group of China Obesity Task Force. [Body mass index reference norm for screening overweight and obesity in Chinese children and adolescents]. Zhonghua Liu Xing Bing Xue Za Zhi. 2004, 25, 97-102.

Did the questions about fruit consumption also include juices? To my knowledge, children have a greater problem with low vegetable consumption, so why there are fruits that children eat more often. This can be mentioned in the introduction.

Response: We truly appreciate your comments. The questions about fruit consumption did not include juices, in this study it only represented fresh fruit consumption.

We truly agree with your opinion that children have a greater problem with low vegetable intake. Actually, it was reported that children and adolescents also have problems with inadequate fresh fruit intake. In addition, the beneficial health effects of fruit are well established[1]. However, in contrast to fruit intake, there is no significant association between the intake of vegetables and hypertriglyceridemia[2, 3]. Notably, low fruit consumption is considered to be the fifth leading contributor to the global disease burden, and thus one of the major attributable risk factors for diseases such as being overweight, hyperglycemia, and hypercholesterolemia[4]. What’s more, fruit also contains a large amount of simple sugars (glucose, fructose, sucrose, etc.), which are well known to induce obesity. Thus, considering the amount of simple sugars found in fruit, it is reasonable to expect that their consumption should contribute to obesity rather than weight reduction. For this reasons, this study focused on the impact of fruit consumption on lipid profile.

According to your suggestion, we revised the Introduction and added some descriptions as follows:

Among the multiple diet factors, vegetable and fruits are important sources of healthy diet. Notably, low fruit consumption is considered to be the fifth leading contributor to the global disease burden[7]. Growing observational evidence suggested that the fruit consumption might parallel the decrease in the risks of obesity, diabetes mellitus, and cardiovascular events in both childhood and adulthood[8, 9]. Pan and his colleagues found a significant inverse association between healthy eating index score of fruits and the risk of metabolic syndrome (MS) among US adolescents, suggesting that a fruit-rich diet could exert beneficial effects in prevention of MS[10]. However, inadequate fruit consumption of children and adolescents worldwide despite the generally higher preference for consumption of fruits than vegetables[11], interventions to encourage fruit consumption during childhood and adolescence might therefore be an effective strategy in reducing disease burden. (line 56-68)

......

In addition, fruit also contains a large amount of fructose, the accumulation of which are detrimental. whether a causal link exists between natural sources of fructose present in fruits and the development of lipid disorders continues to be contested[17]. (line 84-87)

Reference:

[1] Liu RH. Health-promoting components of fruits and vegetables in the diet. Adv Nutr. 2013 May 1;4(3):384S-92S.

[2] Du H, Li L, Bennett D, Guo Y, Key TJ, Bian Z, Sherliker P, Gao H, Chen Y, Yang L, Chen J, Wang S, Du R, Su H, Collins R, Peto R, Chen Z; China Kadoorie Biobank Study. Fresh Fruit Consumption and Major Cardiovascular Disease in China. N Engl J Med. 2016 Apr 7;374(14):1332-43.

[3] Yuan C, Lee HJ, Shin HJ, Stampfer MJ, Cho E. Fruit and vegetable consumption and hypertriglyceridemia: Korean National Health and Nutrition Examination Surveys (KNHANES) 2007-2009. Eur J Clin Nutr. 2015 Nov;69(11):1193-9.

[4] Lim SS, Vos T, Flaxman AD, et al. A comparative risk assessment of burden of disease and injury attributable to 67 risk factors and risk factor clusters in 21 regions, 1990-2010: a systematic analysis for the Global Burden of Disease Study 2010. Lancet. 2012 Dec 15;380(9859):2224-60.

Reference in the manuscript:

[7] LIM SS, VOS T, FLAXMAN AD, DANAEI G, SHIBUYA K, ADAIR-ROHANI H, AMANN M, ANDERSON HR, ANDREWS KG, ARYEE M, et al. A comparative risk assessment of burden of disease and injury attributable to 67 risk factors and risk factor clusters in 21 regions, 1990-2010: a systematic analysis for the Global Burden of Disease Study 2010. Lancet 2012, 380, 2224-2260.

[8] VAN DUYN M, PIVONKA E. Overview of the health benefits of fruit and vegetable consumption for the dietetics professional: selected literature. J Am Diet Assoc 2000, 100, 1511-1521.

[9] LEDOUX T, HINGLE M, BARANOWSKI T. Relationship of fruit and vegetable intake with adiposity: a systematic review. Obes Rev 2011, 12, e143-150.

[10] PAN Y, PRATT C. Metabolic syndrome and its association with diet and physical activity in US adolescents [J]. J Am Diet Assoc 2008, 108, 276-286; discussion 86.

[11] ROSI A, PAOLELLA G, BIASINI B, SCAZZINA F; SINU Working Group on Nutritional Surveillance in Adolescents. Dietary habits of adolescents living in North America, Europe or Oceania: A review on fruit, vegetable and legume consumption, sodium intake, and adherence to the Mediterranean Diet. Nutr Metab Cardiovasc Dis 2019, 29, 544-560.

[17] SHARMA SP, CHUNG HJ, KIM HJ, HONG ST. Paradoxical Effects of Fruit on Obesity. Nutrients 2016, 8, 633.

In verses 74-75 you wrote: "... although there was no consensus on the daily consumption of fruit by children and adolescents" and in 127-129: "According to the 2016 Dietary Guidelines for Chinese School Children [21], inadequate daily intake fruit is defined as eating less than 150 grams per day, and the recommended daily intake is around 300 grams for the pediatric population ", and in conclusion, you repeated this recommendation. Please add an explanation of what is new from your study.

Response: Thanks for your constructive advice. First, we highlighted in the Introduction that “while there was no consensus regarding daily fruit consumption for children and adolescents, especially aimed at reducing adverse lipid profiles.” (line 79-81)

In detail, “According to the Dietary Guidelines for Chinese School-age Children 2016[24], inadequate daily fruit intake was defined as eating fruits less than 150 grams each day, and the recommended daily intake was approximately 250-350 grams for pediatric population.” (line 139-142)

Besides, this manuscript aimed at lipid health of children and adolescents, since children might be more sugar-sensitive than adults, our findings indicated that moderate fruit intake, rather than increased fruit consumption, could be beneficial to lipid health among children and adolescents. According to your suggestion, we revised the Conclusion as follows:

Given the reason that the basis of dietary intake recommendations is related with its health implications, moderate fruit consumption of approximately 150-300 grams each day for 6-7 days/week could be beneficial to lipid health for children and adolescents. (line 421-424)

Also, we emphasized the conclusion in the Abstract:

Moderate fruit consumption was associated with lower odds of lipid disorders, predominantly in girls, younger participants, and those came from higher-educated families. These findings supported the health effect of moderate fruit intake frequently to improve the childhood lipid profiles. (line 30-33)